# Effectiveness of Prophylactic Human Papillomavirus Vaccine in the Prevention of Recurrence in Women Conized for HSIL/CIN 2-3: The VENUS Study

**DOI:** 10.3390/vaccines10020288

**Published:** 2022-02-14

**Authors:** Andrea Casajuana-Pérez, Mar Ramírez-Mena, Estefanía Ruipérez-Pacheco, Inés Gil-Prados, Javier García-Santos, Mónica Bellón-del Amo, Juan J. Hernández-Aguado, Jesus de la Fuente-Valero, Ignacio Zapardiel, Pluvio J. Coronado-Martín

**Affiliations:** 1Department of Obstetrics and Gynecology, University Hospital 12 de Octubre, 28041 Madrid, Spain; 2Gynecology Oncology Unit, Institute of Women’s Health, San Carlos Clinical Hospital (IdISSC), Complutense University, 28040 Madrid, Spain; marramirezmena@gmail.com (M.R.-M.); estefania.ruiperez@salud.madrid.org (E.R.-P.); inesgilpra@gmail.com (I.G.-P.); javiergsantos@yahoo.es (J.G.-S.); monica.bellon@salud.madrid.org (M.B.-d.A.); pluviojesus.coronado@salud.madrid.org (P.J.C.-M.); 3Service of Obstetrics and Gynecology, Infanta Leonor Hospital, Complutense University, 28040 Madrid, Spain; jjhernandeza@salud.madrid.org (J.J.H.-A.); delavalero@gmail.com (J.d.l.F.-V.); 4Gynaecologic Oncology Unit, La Paz University Hospital-IdiPAZ, 28046 Madrid, Spain; ignaciozapardiel@hotmail.com

**Keywords:** HPV vaccine, conization, subsequent disease, persistent/recurrent disease, HSIL/CIN 2-3, cervical cancer prevention

## Abstract

*Background:* Recent data have shown that the human papillomavirus (HPV) vaccine could impact on a decrease in high-grade cervical intraepithelial lesions (HSIL) in women who have undergone surgical treatment. This study aimed to evaluate the efficacy of human papilloma virus (HPV) vaccination against persistent/recurrent disease in patients undergoing conization for high-grade squamous intraepithelial lesion/cervical intraepithelial neoplasia-grade 2-3 (HSIL/CIN 2-3). *Methods:* From January 2009 to March 2019, 563 patients with HSIL/CIN 2-3 underwent conization. The population was divided into two groups according to vaccination status: vaccinated-group (V-Group) and non-vaccinated-group (NV-Group). Bivalent or quadrivalent vaccines were administered indiscriminately. A follow-up was scheduled every 6–12 months according to clinical guidelines. The mean follow-up was 29.6 vs. 36.5 months in the V-group and NV-group, respectively. *Results:* 277 (49.2%) women were vaccinated, while 286 (50.8%) were not. Overall, persistent/recurrent HSIL/CIN 2-3 was presented by 12/277 (4.3%) women in the V-Group and 28/286 (9.8%) in the NV-Group (HR: 0.43, 95% Confidence interval 0.22–0.84, *p* = 0.014). Vaccination was associated with a 57% reduction in HSIL persistence/recurrence after treatment. When no disease was present in the first 6-month follow-up visit, persistence/recurrence rates were very low in both groups: 1.1% in the V-Group vs. 1.5% in the NV-Group (*p* > 0.05). The factor associated with a high-risk of HSIL persistence/recurrence was the presentation of a positive co-test in the first control after treatment (*p* < 0.001). *Conclusions:* Our results corroborate the benefit of HPV vaccination in woman treated for HSIL/CIN 2-3, showing a reduction of persistent/recurrent HSIL/CIN 2-3.

## 1. Introduction

Persistent High Risk-Human Papillomavirus (HR-HPV) infections are strongly associated with the development of High-grade Squamous Intraepithelial Lesion/Cervical Intraepithelial Neoplasia (HSIL/CIN) 2-3 [1,2]. Prophylactic vaccination against HPV with either of any type of the vaccine (bivalent, quadrivalent, nonavalent) has been demonstrated to be the best primary prevention strategy against this disease [3,4].

Patients who have received excisional therapy (conization) for HSIL/CIN 2-3 have a 0.4–19% risk of lesion recurrence [5]. In addition, these women have a 5–10-fold higher risk of developing cervical cancer over the following 10–20 years compared with the general population [6,7]. There is a growing body of evidence showing that prophylactic vaccination against HPV in women who have been treated for Squamous Intraepithelial Lesions (SIL)/CIN may result in a reduction in subsequent disease during follow-up [8,9,10,11]. In addition, it seems that, immunologically, women undergoing a Loop Electrosurgical Excision Procedure (LEEP) showed some changes in inflammatory response in the cervix, reducing TNFa and pro-inflammatory cytokines because surgical intervention eliminates the lesion persistently infected with HPV [12,13]. The anti-inflammatory microenvironment disadvantages a persistent HPV infection. Therefore, if the vaccine were applied at this time, new or recurrent HPV infections could be prevented [14].

In April 2015, the Community of Madrid pioneered an HPV vaccination strategy in patients under 45 years of age with a diagnosis of HSIL/CIN 2-3 who had undergone conization in the previous 3 years. The Gynaecology Oncology Department of the San Carlos Clinical Hospital was among the first centers to benefit from initiating prophylactic vaccination against HPV in treated women. In the last 5 years, different Spanish regions have incorporated free prophylactic vaccination against HPV for women treated for HSIL/CIN 2-3. Since October 2018, the Ministry of Health has supported the Adult Vaccination Calendar for risk groups [15]. In this calendar, prophylactic vaccination against HPV is included for certain risk groups, including women treated for HSIL/CIN 2-3 regardless of age. This strategy had already been implemented in the Community of Madrid in March 2019.

The primary objective of this study was to evaluate the efficacy of HPV vaccination against persistent/recurrent disease (subsequent disease) in patients undergoing conization for HSIL/CIN 2-3 in a real-life setting. For this purpose, a group of treated and vaccinated women were compared with an unexposed group of treated and non-vaccinated women. Secondary objectives were: (1) to analyze the risk of subsequent disease in women with a negative co-test at the 6-month follow-up visit after treatment, and (2) to evaluate the predictive factors of lesion recurrence after excisional therapy for HSIL/CIN 2-3.

## 2. Materials and Methods

### 2.1. Study Design

A retrospective cohort study of women treated by excisional therapy for HSIL/CIN 2-3 was carried out at Hospital Clínico San Carlos between 2009–2019.

The study consecutively included women who met the following inclusion criteria and none of the exclusion criteria:

Inclusion criteria: (1) women older than 18 years who had received excisional therapy for histologically confirmed HSIL/CIN 2-3; (2) women who provided signed informed consent; and (3) women attending the first post-surgery control.

Exclusion criteria: (1) women who do not wish to or cannot provide informed consent; (2) women who do not comply with the study requirements; (3) conized women with lesions other than HSIL/CIN 2-3 (cervix carcinoma, persistent LSIL/CIN 1 with lesion ≤ LSIL/CIN 1 in cone specimen); and (4) women with immunosuppression. All the women included, regardless of the group analyzed, were diagnosed by the same medical team and managed according to the local clinical guidelines adapted from the Spanish Association of Cervical Pathology and Colposcopy (AEPCC) and the *Spanish Society of Gynaecology and Obstetrics* (SEGO) [4,7,16].

Patients diagnosed with HSIL/CIN 2-3 were referred to HPV vaccination. As the vaccine began to be funded in 2015, most women diagnosed from 2009–2014 were not vaccinated, while those diagnosed from 2015–2019 were. Women who had received at least one dose of the HPV vaccine were considered as vaccinated.

### 2.2. Sample Study

Liquid-based cytology and HPV testing were performed in cervical samples that had been previously collected using a cytobrush and stored in PreservCyt solution. After preparing cytology in slides, it was stained with the Papanicolaou method. The sample was classified using the Bethesda 2014 nomenclature.

The detection of HPV was performed by liquid-based cytology using the CLART TEST PAPILLOMAVIRUS HUMAN test that detects the presence of the HPV virus (6, 11, 16, 18, 26, 31, 33, 35, 39, 40, 42, 43, 44, 45, 51, 52, 53, 54, 56, 58, 59, 61, 62, 66, 68, 70, 71.72, 73, 81, 82, 83, 84, 85 and 89). The test has a diagnostic sensitivity and specificity of 98.2% and 100%, respectively, and an analytical specificity of 100%. The analytical sensitivity of this test is 100% when the number of copies is 1000 or 10,000 depending on the type of HPV. The laboratory carried out an external quality control according to the Spanish Society of Pathological Anatomy (SEAP) [17].

### 2.3. Pre-Surgical Evaluation

Patients with an abnormal cytological result and/or a positive HPV test were referred to the Gynaecology Oncology Unit of our hospital for evaluation according to the Cervical Cancer Prevention Guidelines. Patients diagnosed with HSIL/CIN 2-3 or persistent LSIL/CIN 1 underwent cervical conization. Patients who met the inclusion criteria were subsequently enrolled in the study. Women who attended follow-up visits signed the informed consent approved by the local Ethics Commission (24/09/20; 19/320-E).

### 2.4. Surgical Treatment

Cervical LEEP was carried out by minor outpatient surgery under local analgesia and sedation. After infiltration with a local anesthetic and vasoconstrictor in the 4 quadrants of the cervix, conization with a diathermic loop was performed under direct colposcopy vision. The size of the loop varied according to the type of conization and the characteristics of the lesion. Immediate endocervical curettage was performed in all cases as well as selective coagulation of the bleeding areas. Cervical specimens were anatomically oriented, placed in formaldehyde, and sent to Pathological Anatomy for study. All the procedures were performed by members of the Gynaecology Oncology Department, who are experts in the LEEP technique.

### 2.5. Clinical Follow Up

The first assessment was made 30 days after surgery to review the pathology results. Positive margins were diagnosed when a SIL/CIN of any grade involved any edge, or the endocervical curettage was positive.

According to the follow-up protocol of the 2014 Cervical Cancer Prevention Clinical Guide of the AEPCC prevailing at the time of the study, in cases of non-involvement of margins and negative endocervical curettage, the first post-surgery control was carried out with co-testing determination 6 months after conization. In the case of positive endocervical curettage or margin involvement, the first post-surgery control was at 4 months with co-testing, endocervical study, and colposcopy. Subsequently, a new co-test was carried out at 12, 24, and 36 months after the first post-surgery control in patients with previous negative results. After 3 years of negative testing, the patient was returned to the current cervical cancer-screening program. In women with abnormal co-testing at six months, follow-up varied according to the results and colposcopy findings.

The 6-month follow-up visit was defined according to the co-testing result as: (1) persistent lesion (presence of any lesion by cytology or biopsy); (2) persistent HPV infection (presence of HR-HPV); or (3) absence of disease (negative cytology/biopsy and negative determination for HR-HPV).

The clinical outcome of the women at the end of the follow-up was categorized as follows: (1) persistence/recurrence: positive HR-HPV test result and/or presence of histological or cytological SIL/CIN of any grade at a cervical or vaginal location; (2) no disease: negative HR-HPV test and negative Pap test/biopsy. ASCUS lesion is only considered if it is accompanied by HR-HPV infection.

Follow-up was defined as the time from conization to the diagnosis of persistence/recurrence or to the last recorded visit.

### 2.6. Vaccination Status

In 2013, vaccination began to be recommended in patients treated for HSIL/CIN 2-3, although vaccination funding was not included in the national system. Madrid was one of the first regions to support funded vaccination for these patients in 2015, with patients diagnosed with HSIL/CIN 2-3 being referred to their primary care center for vaccination. The bivalent vaccine [2v-HPV] (against genotypes 16/18) or quadrivalent vaccine [4v-HPV] (against genotypes 6, 11, 16 and 18) was administered indiscriminately, depending on the funding policies prevailing at that time. Patients vaccinated with at least one dose of any of the available vaccines were considered as cases. The 2v vaccine had a vaccination schedule of 0–1–6 months, while the 4v was administered at 0–2–6 months.

The vaccination status at conization (defined by the date of administration of the first dose close to the treatment date), as well as the dates of administration of each dose, were collected from the patients’ records.

### 2.7. Ethical Approval

The study was carried out according to the Guidelines to Good Clinical Practice of the International Conference on Harmonization and the Declaration of Helsinki. Enrolled patients signed an informed consent during follow-up visits. The study was carried out according to the clinical protocols of the hospital, without any additional procedures. The study was approved by the Ethics Committee of our institution (Number 19/320-E).

### 2.8. Data Analysis

Quantitative variables are expressed with their mean and standard deviation (SD). Variables showing an asymmetric distribution were expressed with the median and interquartile range (IQR). The association of qualitative variables between the two study groups was compared using the χ2 or Fisher exact test. Quantitative variables were compared by the Student’s T or the Mann-Whitney U test according to the distribution.

The Kaplan-Meier method was used to estimate the distribution of persistence/recurrence in the study groups. A univariate Cox regression analysis was used to find the variables associated with HSIL/CIN 2-3 persistence/recurrence. For all tests, a *p* value less than 0.05 was considered as significant. The data were analyzed with IBM-SPSS software 21.0.

## 3. Results

### 3.1. Population Characteristics

From January 2009 to January 2019, 563 patients were evaluable and included in the study (See Figure 1).

Table 1 shows the characteristics of the study population. The 33 patients treated for persistent LSIL/CIN1 presented a final histological result of HSIL/CIN 2-3 in the cone specimen. A negative result in the conization specimen was found in 34 patients treated by punch biopsy of HSIL/CIN 2-3: 26 were from biopsies with HSIL/CIN2 and 8 with HSIL/CIN3 results. A final diagnosis of LSIL/CIN 1 was found in the cone specimen in 61 women: 49 had had a previous HSIL/CIN2 biopsy and 13 HSIL/CIN3. All these cases were considered as HSIL/CIN 2-3 cases in the analysis.

Positive margins in the cone specimen were found in 131 women (23.4%): 72 involved only endocervical margins, 42 only exocervical, and 17 involving both margins. Of these 131 women, 29 underwent a second conization because of involvement of the endocervical margin, and only one woman continued to show positive margins in the re-conization specimen, being considered as a positive margin for the follow-up. Finally, 103 (18.3%) women were considered as having positive margins during the follow-up period.

### 3.2. Characteristics of Vaccinated and Non-Vaccinated Women

Overall, 277 (49.2%) were vaccinated and 286 (50.8%) were not. The mean age was 36.91 years (SD 8.15). Table 1 shows the clinical characteristics, the histological diagnosis of the conization specimen, and the margin status of the vaccinated and non-vaccinated women. No differences were found between the two groups.

### 3.3. Vaccination Scheme

Of the 277 vaccinated women, 255 (92.1%) received the three doses, 16 (5.8%) received two doses, and 6 (2.2%) received only one dose.

Among the vaccinated women, 67 (25.1%) received the first dose prior to conization: 12 (17.9%) received the first dose within 30 days before the intervention, 42 (62.7%) between 1 to 9 months, and 13 (19.4%) more than 9 months before conization.

Two hundred women were vaccinated after treatment: 14 (7%) received the first dose the first month after conization, 135 (67.8%) at between one and six months, and 51 (25.12%) were vaccinated six months after treatment. Ten women were vaccinated during follow-up, but the dates were not reported.

### 3.4. Results at 6-Month Follow-Up Visit

The mean time from treatment to the first visit was 7.1 months (SD 8). The status at this control is shown in Table 2. Of 167 patients with HR-HPV infection, 106 (63.5%) had normal cytology. The 10 women with persistent HSIL in the first post-conization visit underwent a second treatment after a HSIL/CIN 2-3 target biopsy.

### 3.5. Clinical Outcome at the End of Follow-Up

The mean follow-up was 33.1 months (SD 17.60; median 32.9 months). For the vaccinated women, the mean follow-up was 29.6 months (SD 15.2; median 26.5 months), being 36.5 (SD 17.9; median 36.6 months) (*p* < 0.001) for the non-vaccinated women.

At the end of follow-up, 55 of 563 (9.8%) women had persistent/recurrent HPV 16–18 infection. Persistent/recurrent infection was found in 28 of 286 (9.7%) women in the non-vaccinated group and in 27/277 (9.74%) women in the vaccinated group (*p* = 0.814). Figure 1 summarizes the distribution of patients and final outcome by time period (2009–2014 and 2015–2019). There was a tendency to lower HSIL/CIN 2-3 recurrence in the vaccinated group in comparison with the non-vaccinated group (See Figure 2).

Table 3 12/277 (4.3%) vs. 28/286 (9.8%), *p* < 0.05). No differences were found when comparing HSIL/CIN 2-3 between the two groups considering only patients with no disease at the 6-month follow-up visit.

Considering only HSIL/CIN 2-3 recurrence associated with the vaccine genotype (16/18), a lower non-significant recurrence rate was found in vaccinated women compared to non-vaccinated women: 8/277 (2.9%) vs. 11/286 (3.8%); *p* = 0.529.

However, the two vaccinated women with HSIL/CIN 2-3 recurrence showed another genotype not included in the HPV vaccines: both women received the first dose of the vaccine one month after conization. The recurrence appeared 3 years later carrying an HPV 58 in the first woman and 2 years later carrying an HPV 33 in the second woman. On the other hand, when no disease was detected at the 6-month follow-up visit, the risk of recurrent HPV 16/18 infection decreased from 48/184 (26.1%) to 7/379 (1.8%) (*p* < 0.001). No differences were found in relation to recurrent HPV-16/18 infection between the vaccinated and non-vaccinated groups (4/184 (2.2%) vs. 3/195 (1.5%), respectively).

Table 4 shows the univariate analysis for persistent/recurrent HSIL/CIN 2-3 at the end of follow-up. The factor associated with a low risk of persistent/recurrent HSIL/CIN 2-3 was the HPV vaccination status (*p* = 0.014), and the factor associated with a high risk of persistent/recurrent was the presence of a positive co-test at 6 months after conization (*p* < 0.001).

## 4. Discussion

To our knowledge, this is the largest Spanish study to evaluate the effect of the HPV vaccine in women treated for HSIL/CIN 2-3 in clinical practice with the same follow-up protocol. The results of our study support the use of HPV vaccination to reduce the risk of developing persistent/recurrent HSIL/CIN 2-3 after surgical treatment.

Several studies [8,9,11] have shown that HPV vaccination in women undergoing conization for HSIL/CIN 2-3 can reduce the risk of developing subsequent disease. The SPERANZA was the first prospective study evaluating the clinical effectiveness of the HPV vaccine in reducing HSIL/CIN 2+, with a recurrence risk reduction of 81.2%. In our study, although the HSIL/CIN 2-3 rate was significantly lower in the vaccinated group, we found a risk reduction of 57%, similar to the 59% published in the meta-analysis by Jentschke [14]. Gomez de la Rosa et al. reported an intermediate relative risk of 73.5% [18]. The protective effect of the HPV vaccine was particularly clear against HSIL/CIN 2-3 lesions, since HPV 16/18 was present in only 20% of low-grade lesions. Our LSIL/CIN 1 rates were similar to those of del Pino et al. [11], with no differences between the vaccinated and non-vaccinated groups.

While it has been hypothesized that the HPV vaccine could evoke local antibodies that prevent virus entrance in the basal layer [8], the effectiveness of prophylactic HPV vaccination against prevalent infection has not been demonstrated [19,20,21]. Our data suggest that the HPV vaccine does not reduce the risk of persistent/recurrent HPV infection (neither genotype 16/18) (*p* = 0.814), according to the findings of del Pino et al. [11]. However, our study found a lower rate of high-grade lesional recurrence associated with the HPV 16–18 genotypes in the vaccinated group, albeit without significant differences between groups. This result may be due to the low number of affected patients and would probably be considered more significant with a larger patient sample. Therefore, recurrence associated with vaccine genotypes only should be studied in greater depth.

Persistent HSIL at the 6-month follow-up visit was similar in both groups (4/277 [1.4%] in the vaccinated vs. 6/286 [2.1%] in the non-vaccinated group). This could be explained, first of all, because the vaccine does not have a therapeutic but a prophylactic effect, and, in the second place, due to the fact that antibody titres have a maximum peak 7 months after the administration of the first dose [22,23] and most of our patients received the first dose of the HPV-vaccine after conization; thus, vaccination was not completed until 6 months after treatment. On the other hand, data available show excellent duration of the protection for the time periods through which they have been studied. Persistent antibody levels and protection against HPV infection have also been reported up to 14 years following vaccination, suggesting no need for a booster dose during that period [24]. However, the precise level of antibodies required for protection against infection is unknown, so further studies are necessary. In this regard, our results show that the HPV vaccine reduces lesion recurrence at least 5 years after administration. No-disease at the 6-month follow-up visit is related to a decrease in lesion recurrence and HPV infection, however, although in the future a different follow-up strategy could be proposed for vaccinated patients with a negative cotest at 6 months post-treatment; to date, there is not enough scientific evidence to make changes to the clinical guidelines. Interestingly, in the present series, all the women showed a very low rate of recurrent HPV infection and recurrent HSIL/CIN 2-3 regardless of vaccination status.

The strength of our study includes the large size of the cohort and the long follow-up period following the same protocol. In addition, our study showed low rates of positive margins and positive co-tests at the 6-month follow-up visit, according to the quality standards of the Colposcopy Unit. Correct patient treatment could justify the low percentage of persistent/recurrent lesions regardless of vaccination status. On the other hand, some limitations must be taken into account: women were vaccinated at different time points, not all women received the full vaccination schedule, and, due to the retrospective nature of the study, the vaccination type (2v-HPV or 4v-HPV) could not be analyzed. In addition, we cannot be sure that no women returned to the screening program, presented late recurrence, and were evaluated another hospital. Finally, another limitation is that the mean length follow-up was shorter in the vaccinated group because very few women were vaccinated before 2015.

## 5. Conclusions

In conclusion, this study corroborates the benefit of HPV vaccination in women treated for HSIL/CIN 2-3, showing that HPV vaccination is associated with a reduction in the risk of persistent/recurrent HSIL/CIN 2-3 after conization and that the factor associated with a high risk of persistent/recurrent HSIL/CIN 2-3 is to have abnormal pap-smear results and/or a positive HR-HPV genotype at the first control after treatment.

## Figures and Tables

**Figure 1 vaccines-10-00288-f001:**
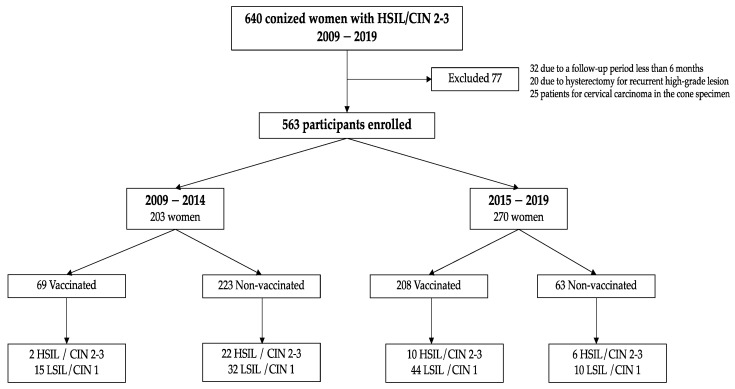
Participant disposition. Abbrevations: CIN: Cervical Intraepithelial Neoplasia. HSIL: High-grade Squamous Intraepithelial Lesion. LSIL: Low-grade Squamous Intraepithelial Lesion.

**Figure 2 vaccines-10-00288-f002:**
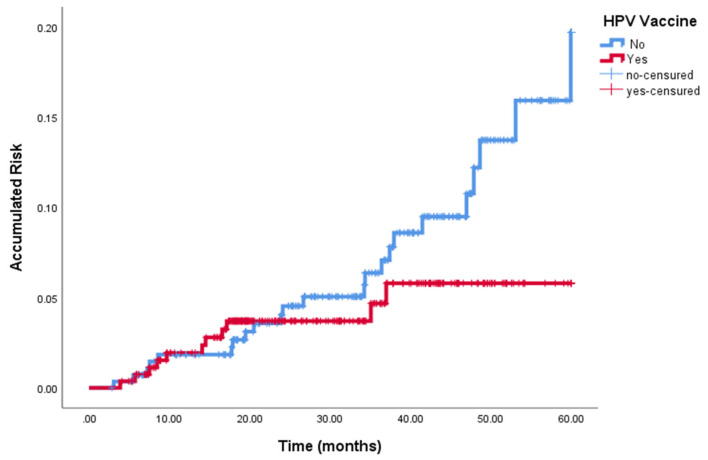
Persistent /Recurrent HSIL/CIN 2-3 in vaccinated and non-vaccinated women.

**Table 1 vaccines-10-00288-t001:** Clinical characteristics and data referring to conization in vaccinated and non-vaccinated women. CIN: Cervical Intraepithelial Neoplasia. HPV: Human Papilloma Virus. HR: High Risk HSIL: High-grade Squamous Intraepithelial Lesion. LSIL: Low-grade Squamous Intraepithelial Lesion.

Clinical Characteristics	Vaccinated Group (%) (*n* = 277)	Non-Vaccinated Group (%) (*n* = 286)	*p*
Age			0.060
<35 years	128 (46.2)	109 (38.2)	
≥35 years	149 (53.8)	176 (61.8)	
Smokers			0.077
No	122 (44)	118 (41.3)	
Yes	61 (22)	47 (16.4)	
Unknown	94 (34)	121 (42.3)	
History of pregnancies			0.079
No	153 (55.4)	133 (46.5)	
Yes	123 (44.6)	150 (55.4)	
Unknown	1(0.3)	3 (1.1)	
HPV Genotype			0.187
16/18 HPV	66 (23.8)	39 (13.7)	
Other HR HPV	51 (18.4)	42 (14.7)	
Non-HR HPV	11 (4.0)	14 (4.9)	
Indication of conization			0.908
Persistent LSIN/CIN1	17 (6)	16 (5.8)	
HSIL/CIN 2-3	266 (94)	261 (94.2)	
Margin status			0.270
Negative	221 (79.8)	239 (83.5)	
Positive	56 (20.2)	47 (16.6)	

**Table 2 vaccines-10-00288-t002:** Status at six months in the vaccinated and non-vaccinated group. HPV: Human Papilloma Virus. HSIL: High-grade Squamous Intraepithelial Lesion. HR: High Risk. LSIL: Low-grade Squamous Intraepithelial Lesion.

Status at 6 Months	Vaccinated Group (%) (*n* = 277)	Non-Vaccinated Group (%) (*n* = 286)	*p*
Cytology			0.490
Negative	239 (86.3)	246 (86)	
LSIL/ASCUS	34 (12.3)	34 (12.3)	
HSIL	4 (1.4)	6 (2.1)	
HR-HPV testing			0.707
Positive	83 (30)	84 (29.4)	
Negative	194 (70)	202 (70.6)	
Cotesting			0.657
Negative	184 (66.4)	195 (68.2)	
Positive ^1^	93 (33.6)	91 (31.8)	

^1^ Positive co-testing indicates the presence of HR-HPV or abnormal cytology.

**Table 3 vaccines-10-00288-t003:** Distribution and final clinical outcome of all women included in the study (**A**), and only women with no disease at 6-month follow-up visit (**B**). CIN: Cervical Intraepithelial Neoplasia HSIL: High Squamous Intraepithelial Lesion. LSIL: Low Squamous Intraepithelial Lesion; OR: odds ratio, CI: confidence interval.

**(A) 563 Conized Women with HSIL/CIN 2-3**			
	**Vaccinated Women** **277 (49.2%)**	**Non-Vaccinated Women** **286 (50.8%)**	**OR (95% CI)**
No subsequent disease	206 (74.4%)	216 (75.5%)	1
Persistent/Recurrent	59	42	1.5 (0.9–2.3)
LSIL/CIN 1	(21.3%)	(14.7%)	
Persistent/Recurrent	12	28	0.4 (0.2–0.9) ^1^
HSIL/CIN 2-3	(4.3%)	(9.8%)	
**(B) 379 Women Treated for HSIL/CIN 2-3 with No Disease at 6-Month Follow-Up Visit**			
	**Vaccinated Women** **184 (48.5%)**	**Non Vaccinated Women** **195 (51.4%)**	**OR (95% CI)**
No subsequent disease	161 (87.5%)	182 (93.3%)	1
Persistent/Recurrent	21	10	2.4 (1.1–5.2) ^1^
LSIL/CIN 1	(11.4%)	(5.1%)	
Persistent/Recurrent	2	3	0.7 (0.1–4.5)
HSIL/CIN 2-3	(1.1%)	(1.5%)	

^1^*p* < 0.05.

**Table 4 vaccines-10-00288-t004:** Univariate Cox regression analysis of risk factors associated with persistent/recurrent HSIL. SIL: squamous intraepithelial lesion; HR: Hazard ratio; CI: confidence interval; HR- HPV: high risk—human papillomavirus.

Variable	Persistent/Recurrent HSIL/CIN 2-3 at the End of Follow-Up (Univariate Analysis)		
	HR	95% CI	*p*
Age			0.281
<35 years	1		
≥35 years	0.70	0.37–1.37	
Currently Smoke			0.367
No	1		
Yes	0.64	0.25–168	
Presence of HR-HPV			0.340
No	1		
Yes	0.44	0.08–2.35	
Margin in conization specimen			0.475
Negative	1		
Positive	1.30	0.63–2.70	
Vaccination Status			0.014
No	1		
Yes	0.42	0.21–0.84	
6-month follow-up visit			0.001
Negative co-test			
Positive co-test	22.62	7.91–64.67	

## Data Availability

The datasets generated and analyzed during the current study are available from the corresponding author upon reasonable request.

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
