# Peer review of "Effectiveness of Prophylactic Human Papillomavirus Vaccine in the Prevention of Recurrence in Women Conized for HSIL/CIN 2-3: The VENUS Study"

_vaccines, 2022, doi:10.3390/vaccines10020288_

Round 1

Reviewer 1 Report

The authors describe the outcomes of women after conization for HSIL with and without HPV vaccination in a large cohort. The article is well written, and should be of interest to the general reader.

There are a few points which need adaptation and/or clarification.

Formatting error:

First, there seems to be a formatting error (see l. 186, 225, 244, 247, 247, and 272).

Definition of positive margin (l. 126-127) :

The authors define a positive margin as « a SIL/CIN of any grade .. either close to or involved any edge, or the endocervical curettage was positive.»

This is highly unusual – why do the authors mean with «close to the edg » ? What is the reason for this definition ?  What happens if these «close to the edge» cases are not counted as a positive margin?

Definition of « prophylactic » vaccination (e.g. l. 2 and several others) :

The authors label the vaccines in this study as « prophylactic ». However, as the vaccination is assessed for its potential to influence the course of persistent disease, it should rather be labeled as prophylactic and therapeutic.

Definition of great involvement (l. 197):

The authors say that «29 (patients) underwent a second conization because of great involvement of the endocervical margin». What is the definition of «great» involvement ? This terminlogy is rather unusual. Please clarify.

Definition of an abnormal co-testing result (e.g. l. 139/202):

The authors define an abnormal result at co-testing as «at least ASCUS and/or HR-HPV positivity». I wonder how many cases were found with ASCUS and HPV negativity – these cases may by some be regarded as «normal», i.e. low likelihood of HPV-related abnormality.

Sentence not clear (l. 263) :

I think there is something missign/spelling mistake in the sentence « The recurrence appeared 3 years later carrying a HPV 263 In the first women and 2 years later carrying a HPV 33. Should it rather be « The recurrence appeared 3 years later carrying a HPV 263 in the first women and 2 years later carrying a HPV 33 in the second women ». ? Please clarify.

Sentence not clear (l. 300) :

The following sentence is unclear: «This result may be due to the low number of patients presenting recurrence and, would probably achieve significance with a larger sample size.»

Please re-phrase.

Date of vaccination (line 218) :

The authors state that 13 patients (almost 20%) received the vaccination more than 9 months before conization. What was the reason for the vaccination in these cases (e.g. newly diagnosed HSIL? Prophylactic «routine» vaccination in women without HPV-associated lesions)?

In addition, was there a difference in the cliical course between women who had had their vaccination > 9 months prior to conization and those who received it later?

Table 1:

The authors mention «pregnancies» in table 1. However, it is not obvious what this means. Have the women been pregnant before or during the study? Please clarify.

Conclusion (l. 327 ff) :

Are there any recommendations regarding follow-up based on the findings in the study? It may be argued that vaccinated women with normal results at co-testing 6 months after the conization may be eligible for a less rigid scheme of follow-up tests – I wonder what the authors think about this.

Table 2:

Table 2 mentions normal, LSIL, and HSIL as results categories. However, there is no mentioning of ASCUS. How many cases were there? According to the authors definition, ASCUS was recorded as an abnormal result (see e.g. line 139 or 202).

Author Response

REVIEWER 1

Dear Reviewer,

Thank you very much for your contributions. I will answer point by point each of them with the text in red. Modifications will appear in Word document as tracked changes.

Point 1. Formatting error: First, there seems to be a formatting error (see l. 186, 225, 244, 247, 247, and 272).

RE: sorry for the mistake. The cross reference must have been deleted. I have written it manually so it doesn't happen anymore:

  • The absence of reference alluded to “Table 1”. It is already corrected in l. 209.
  • The absence of reference alluded to “Table 2”. It is already corrected in l. 272.
  • The absence of reference alluded to “Figure 2”. It is already corrected in l. 297.
  • The absence of reference alluded to “Table 3”. It is already corrected in l. 303.
  • The absence of reference alluded to “Table 4”. It is already corrected in l. 340.

Point 2. Definition of positive margin (l. 126-127):

The authors define a positive margin as « a SIL/CIN of any grade... either close to or involved any edge, or the endocervical curettage was positive.»

This is highly unusual – why do the authors mean with «close to the edge»? What is the reason for this definition?  What happens if these «close to the edge» cases are not counted as a positive margin?

RE: Thank you very much for your comment. You are right and it is due to a translation error. We were referring to the lesion contacting the margin at one point or affecting its entirety. However, it was confusing and we have modified it: Positive margins were diagnosed when a SIL/CIN of any grade involved any edge, or the endocervical curettage was positive (l. 141-142).

Point 3. Definition of « prophylactic » vaccination (e.g. l. 2 and several others):

The authors label the vaccines in this study as « prophylactic ». However, as the vaccination is assessed for its potential to influence the course of persistent disease, it should rather be labeled as prophylactic and therapeutic.

RE: I understand your contribution and I agree with it but still, to date, there is not enough evidence to say that the vaccine has therapeutic value. Currently, its prophylactic action is only accepted.

Point 4. Definition of great involvement (l. 197):

The authors say that «29 (patients) underwent a second conization because of great involvement of the endocervical margin». What is the definition of «great» involvement? This terminlogy is rather unusual. Please clarify.

RE: I agree that it is not an appropriate term, thank you. We were referring to patients with endocervical margin affect and positive endocervical curettage. We have removed the word "great". (l. 234).

Point 5. Definition of an abnormal co-testing result (e.g. l. 139/202):

The authors define an abnormal result at co-testing as «at least ASCUS and/or HR-HPV positivity». I wonder how many cases were found with ASCUS and HPV negativity – these cases may by some be regarded as «normal», i.e. low likelihood of HPV-related abnormality.

RE: Certainly, it may not be well understood. Patients with ASCUS and negative HPV were considered as having a normal result. The text has been modified for better understanding in “Materials and Methods” as shown below: ASCUS lesion is only considered if it is accompanied by HR-HPV infection” (l. 171).

Point 6. Sentence not clear (l. 263):

I think there is something missign/spelling mistake in the sentence « The recurrence appeared 3 years later carrying a HPV 263 In the first women and 2 years later carrying a HPV 33. Should it rather be « The recurrence appeared 3 years later carrying a HPV 263 in the first women and 2 years later carrying a HPV 33 in the second women ». ? Please clarify.

RE: You are right, it is a spelling mistake that has been corrected: “The recurrence appeared 3 years later carrying a HPV 58 in the first woman and 2 years later carrying a HPV 33 in the second woman” (l.314).

Point 7. Sentence not clear (l. 300) :

The following sentence is unclear: «This result may be due to the low number of patients presenting recurrence and, would probably achieve significance with a larger sample size.». Please re-phrase.

RE: Thank you. The original sentence has been replaced by the following: “This result may be due to the low number of affected patients and would probably be considered more significant with a larger patient sample.” (L. 382 – 383)

Point 8. Date of vaccination (line 218):

The authors state that 13 patients (almost 20%) received the vaccination more than 9 months before conization. What was the reason for the vaccination in these cases (e.g. newly diagnosed HSIL? Prophylactic «routine» vaccination in women without HPV-associated lesions)? In addition, was there a difference in the clinical course between women who had had their vaccination > 9 months prior to conization and those who received it later?

RE: Thank you for your point. Women who received vaccination more than 9 months before conization was because prophylactic routine or woman’s desire to be vaccinated. Our protocols do not make different follow-up strategies depending on the vaccination status.

Point 9. Table 1:

The authors mention «pregnancies» in table 1. However, it is not obvious what this means. Have the women been pregnant before or during the study? Please clarify.

RE: Thank you, it certainly wasn't clear. It is referred to “history of pregnancy”. It is already modified in Table 1.

Point 10. Conclusion (l. 327 ff):

Are there any recommendations regarding follow-up based on the findings in the study? It may be argued that vaccinated women with normal results at co-testing 6 months after the conization may be eligible for a less rigid scheme of follow-up tests – I wonder what the authors think about this.

RE: Thank you for your input. This is an interesting topic that I have incorporated into the discussion. Although in the future a different follow-up strategy could be proposed for vaccinated patients with a negative co-testing at 6 months control, to date, there is not enough scientific evidence to make changes to the clinical guidelines (l. 406-408).

Point 11. Table 2:

Table 2 mentions normal, LSIL, and HSIL as results categories. However, there is no mentioning of ASCUS. How many cases were there? According to the authors definition, ASCUS was recorded as an abnormal result (see e.g. line 139 or 202).

RE: Thank you, the ASCUS category was counted together with LSIL. I have incorporated it into the Table 2 to avoid confusion as it shown below:

Status at 6 months

Vaccinated group (%)

(n= 277)

Non-vaccinated Group (%)

(n = 286)

p

Cytology

Negative

LSIL/ASCUS

HSIL

239 (86.3)

34 (12.3)

4 (1.4)

246 (86)

34 (12.3)

6 (2.1)

0.490

HR-HPV testing

Positive

Negative

83 (30)

194 (70)

84 (29.4)

202 (70.6)

0.707

Cotesting

Negative

Positive1

184 (66.4)

93 (33.6)

195 (68.2)

91 (31.8)

0.657

We hope that we have provided enough detail for you to reconsider publishing our manuscript, and we remain at your disposal for any further clarification you may require.

Best Regards, 

Andrea Casajuana.

Reviewer 2 Report

Some minor errors were detected at:

Line 186, 225, 244, 247, 272.

Table 1: Age column, vaccinated group:

Is it true the sum up of 46.2% + 53.6% no equally 100%?

The column pregnancies:

Vaccinated group: 153 + 123 not equally to 277

Non vaccinated group: 113 + 150 not equally to 286.

Part 3.3. Vaccination Scheme

Which table(s) represented the explanation on 216 – 222?

Part 3.4. Results at six months control after conization

Which table(s) represented the explanation on 224 – 228?

Table 4. Why was the univariate logistic Cox’s regression analysis used? 

Author Response

REVIEWER 2

Dear Reviewer,

Thank you very much for your contributions. I will answer point by point each of them with the text in red. Modifications will appear in Word document as tracked changes.

Point 1. Abstract. Line 19: please add 1 sentence concisely before “This study aimed to evaluate……”

RE: following your recommendation, the following sentence has been introduced:

“Recent data have shown that human papillomavirus (HPV) vaccine could impact on a decreased of high-grade cervical intraepithelial lesion (HSIL) in women who underwent surgical treatment”. (L. 19-21).

Point 2. Introduction. Line 51-53: please provide 2-3 sentences to comprehend the molecular and immunological mechanisms beyond it.

RE: For a better understanding, following your advice, the text has been modified, expanding the information: “In addition, it seems that immunologically, women undergoing a Loop Electrosurgical Excision Procedure (LEEP) showed some changes in inflammatory response in the cervix, reducing TNFa and pro-inflammatory cytokines because surgical intervention eliminates the lesion persistently infected with HPV. The anti-inflammatory microenvironment disadvantages a persistent HPV infection. Therefore, if the vaccine were applied at this time, new or recurrent HPV infections could be prevented.” (L. 54-59).

Point 3. Materials and Methods.

3.1. Line 135: please provide the detailed and scientific rationale or previous studies that support the duration (12, 24, and 36 months).

RE: revisions post-treatment after a negative first-control are done annually during the next 3 years, following clinical guidelines from Spanish Association of Cervical Pathology and Colposcopy (AEPCC) and Spanish Society of Gynaecology and Obstetrics (SEGO) (l.94; l. 143). You can consult these guidelines in bibliography [4], [7]

3.2. Line 155: If the vaccines were given indiscriminately, how will it affect the patients? Were there studies or previous researches that support it?

RE: Scientific evidence to date has shown an impact in reducing recurrence in women who have received excisional therapy against oncogenic genotypes (16, 18) included in both vaccines. It would have been interesting to analyze if there is a different impact with each vaccine but it is a limitation of our study: “Due to the retrospective nature of the study, the vaccination type (2v-HPV or 4v-HPV) could not be analysed” (l. 417-418).

Point 4. Results.

4.1. Line 186, 225, 244, 247, 272.

RE: Sorry for the mistake. The cross reference must have been deleted. I have written it manually so it doesn't happen anymore:

  • The absence of reference alluded to “Table 1”. It is already corrected in l. 209.
  • The absence of reference alluded to “Table 2”. It is already corrected in l. 272.
  • The absence of reference alluded to “Figure 2”. It is already corrected in l. 297.
  • The absence of reference alluded to “Table 3”. It is already corrected in l. 303.
  • The absence of reference alluded to “Table 4”. It is already corrected in l. 340.

4.2 Table 1: Age column, vaccinated group. Is it true the sum up of 46.2% + 53.6% no equally 100%?

RE: Effectively, the rounding was not done correctly. It is already corrected: 128 (46.2%) + 149 (53.8%) = 277 (100%).

4.3 The column pregnancies: Vaccinated group: 153 + 123 not equally to 277. Non vaccinated group: 113 + 150 not equally to 286.

RE: Thank you, it could certainly look confusing. The explanation is that the History of pregnancies of some of the patients was unknown. To avoid confusion, an "Unknown" row has been added to the table. Reviewing the database, we have noticed a transcription error in the numbers, which we have modified.

The table would look like this:

Clinical Characteristics

Vaccinated group (%)

(n= 277)

Non-vaccinated Group (%)

(n = 286)

p

Pregnancies

No

Yes

Unknown

153 (55.4)

123 (44.6)

1(0.3)

133 (46.5)

150 (55.4)

3 (1.1)

0.079

4.4. Part 3.3. Vaccination Scheme. Which table(s) represented the explanation on 216 – 222?

RE: These results are not represented in any table; they are only described in the text. If you consider that making a table can add value to the article, please let me know.

4.5. Part 3.4. Results at six months control after conization. Which table(s) represented the explanation on 224 – 228?

RE: The cross reference must have been deleted. The referred Table was “Table 2”. It is already corrected in l. 272.

4.6 Table 4. Why was the univariate logistic Cox’s regression analysis used? 

RE: Univariate logistic Cox’s regression analysis was used because a Hazard Ratio was performed to analyse recurrence over time.

Point 5. Discussion.

5.1. Explanation stated on line 295 – 297 were as if contradictory with the conclusion. How were the confirmation from the authors?

RE: line 295-297 (now 377-379) are referring to a reduce in HPV infection but not in HSIL lesions as discussed in the conclusion.

5.2. Line 311: … no need for a booster dose… How was the scientific rationale or mechanisms beyond it?

RE: Although further data will be necessary, to date, the need for a booster cannot be recommended. The requested information has been expanded:

“On the other hand, data available shows excellent duration of the protection for the time periods through which they have been studied. Persistent antibody levels and protection against HPV infection have also been reported up to 14 years following vaccionation, suggesting no need for a booster dose during that period. However, the precise level of antibodies required for protection against infection is unknown so futher studies are necessary.” (l.398 – 403).

Point 6. References. Because there are only 20 references in this manuscript, it is recommended to add with updated references within the last 5 years. There are a total of 12 references that were obsolete, such as: 2001, 2007, 2013, 2014, 2015, and 2016.

RE: Thanks for your input. The bibliography has been updated with more recent articles.

  1. Van Dyne EA, Henley SJ, Saraiya M, Thomas CC, Markowitz LE, Benard VB. Trends in Human Papillomavirus-Associated Cancers - United States, 1999-2015. MMWR Morb Mortal Wkly Rep. 2018 Aug 24;67(33):918-924. doi: 10.15585/mmwr.mm6733a2. PMID: 30138307; PMCID: PMC6107321.

  1. AEPCC-Guía: PREVENCIÓN SECUNDARIA DEL CANCER DE CUELLO DEL ÚTERO, 2022. CONDUCTA CLÍNICA ANTERESULTADOS ANORMALES DE LAS PRUEBAS DE CRIBADO. Coordinador: Torné A. Secretaria: del Pino M. Autores: Torné A; Andía, D; Bruni L; Centeno C; Coronado P; Cruz Quílez J; de la Fuente J; de Sanjosé S; Ibáñez R; Lloveras B; Lubrano A Matías Guiu X; Medina N; Ordi J; Ramírez M; del Pino M.

  1. Cox JT; Palefsky JM. (2021). Human papillomavirus vaccination. In MS Hirsch (Ed.), UpToDate. Retrieved February 10, 2022, from https://www.uptodate.com/contents/human-papillomavirus-vaccination

We hope that we have provided enough detail for you to reconsider publishing our manuscript, and we remain at your disposal for any further clarification you may require.

Best Regards,

Andrea Casajuana.

Reviewer 3 Report

This study appears to be a retrospective secondary analysis of a study that prospectively followed women with undergoing conization for high-grade squamous intraepithelial lesion /cervical intraepithelial neoplasia-grade 2-3 (HSIL/CIN2-3). The objective was to evaluate whether women who received any type of the HPV vaccine after diagnosis and treatment had lower rates of persistence or recurrence of disease. The results of this study highlight the importance of vaccinating older women who are at risk of HPV-related neoplasia, an age-group that would otherwise not routinely receive the HPV vaccine.

The study and results have merit, but I found it somewhat difficult to follow in places. Below are my comments:

  1. Introduction, line 44 ‘infection’ should be ‘condition’ or ‘disease’. I believe the authors meant that HPV vaccination was associated with a reduction in lesions/neoplasia and not HPV infection in the referenced studies. Or perhaps both? Please clarify.
  2. Lines 51-53 are confusing and seems to be contradictory. Please clarify.
  3. Lines 65-66 is a redundant sentence (same information is presented in miles 67-69) and can be removed.
  4. Lines 70-71: This is a cohort study so therefore the unvaccinated group is not a control group. It is the unexposed group. HPV vaccination is the exposure of interest and the participants are followed for the development of a condition of interest (recurrent/persistent HSIL/CIN2-3 or HPV infection).
  5. Throughout: the authors keep referring to the 6-month follow-up visit (or 6-month post treatment visit) as the 6-month control. I suggest changing the terminology to the former as control is used in a different context in research studies.
  6. Lines 109-111 and lines 129-131: The guidelines are from 2014. Which guidelines were used for women diagnosed/enrolled prior to the publication of these guidelines?
  7. The authors refer to this as a retrospective cohort but the description in the methods suggest that it was a prospective cohort (re lines 111-114). Or is this a secondary analysis of another study? If so, what was the primary purpose of the parent study?
  8. Lines 136-138: If women were returned to the cervical cancer screening program after 3 years of negative follow-up results, were any women enrolled more than once? If so, how many? The authors should exclude the subsequent enrollments or use statistical methods that account for the collinearity in the data if women were enrolled more than once.
  9. Line 177: It should be univariate Cox regression analysis.
  10. A flow diagram would clarify the disposition of participants. It is very difficult to follow as written in the text of results. I would also like to see the number enrolled, vaccinated, and with outcomes of interest stratified by time period (2009-2014 and 2015-2019) in a flow diagram.
  11. There are a few instances where the text states Error! Reference source not found. These should be reconciled.
  12. Percentage in Table 1 for HPV genotype are incorrect. Also, are these among HPV positive women? I think the authors should include the category for HPV negative.
  13. Lines 241-242: Why is the denominator for negative pap smear 241 and instead 563?

Author Response

REVIEWER 3

Dear Reviewer,

Thank you very much for your contributions. I will answer point by point each of them with the text in red. Modifications will appear in Word document as tracked changes.

Point 1. Introduction, line 44 ‘infection’ should be ‘condition’ or ‘disease’. I believe the authors meant that HPV vaccination was associated with a reduction in lesions/neoplasia and not HPV infection in the referenced studies. Or perhaps both? Please clarify.

RE: Thank you for your input. I have changed the word “infection” by “disease” (l. 47).

Point 2. Lines 51-53 are confusing and seems to be contradictory. Please clarify

RE: For a better understanding, following your advice, the text has been modified, expanding the information: “In addition, it seems that immunologically, women undergoing a Loop Electrosurgical Excision Procedure (LEEP) showed some changes in inflammatory response in the cervix, reducing TNF-a and pro-inflammatory cytokines because surgical intervention eliminates the lesion persistently infected with HPV. The anti-inflammatory microenvironment disadvantages a persistent HPV infection. Therefore, if the vaccine were applied at this time, new or recurrent HPV infections could be prevented.” (l.54-59).

Point 3. Lines 65-66 is a redundant sentence (same information is presented in miles 67-69) and can be removed.

RE: Thank you for your suggestion, the sentence has been removed.

Point 4. Lines 70-71: This is a cohort study so therefore the unvaccinated group is not a control group. It is the unexposed group. HPV vaccination is the exposure of interest and the participants are followed for the development of a condition of interest (recurrent/persistent HSIL/CIN2-3 or HPV infection).

RE: I agree is not the correct term. I have replaced the word “control” by “unexposed”. It would be as follows: For this purpose, a group of treated and vaccinated women were compared with an unexposed group of treated and non-vaccinated women. (l. 74).

Point 5. Throughout: the authors keep referring to the 6-month follow-up visit (or 6-month post treatment visit) as the 6-month control. I suggest changing the terminology to the former as control is used in a different context in research studies.

RE: thank you for your suggestion. The “6-month control” has been replaced by “6-month follow-up visit” throughout the text.

Point 6. Lines 109-111 and lines 129-131: The guidelines are from 2014. Which guidelines were used for women diagnosed/enrolled prior to the publication of these guidelines?

RE: Before de publication of these guidelines “Progresos Guidelines 2006” were used, without differences in the follow-up strategy. We had only referenced the most recent guidelines, but this one has been added as well as reference 17: “Puig-Tintoré LM, Cortés J, Castellsagué X, Torné A, Ordi J, de San José S, et at. Prevención del cáncer de cuello uterino ante la vacunación frente al virus del papiloma humano. Prog Obstet Ginecol 2006; 46 (Supl 2): 5-62.”

Point 7. The authors refer to this as a retrospective cohort but the description in the methods suggest that it was a prospective cohort (re lines 111-114). Or is this a secondary analysis of another study? If so, what was the primary purpose of the parent study?

RE: This is a retrospective cohort study though the description can be confusing. As it is a retrospective study, Informed Consent was not mandatory. We gave it to the patients as they came to the follow-up visit. Patients referred to the screening program before we started the recruitment do not have signed consent, although all patients evaluated in our unit give their consent for data analysis. It has been clarified in the text: “Patients diagnosed with HSIL/CIN 2-3 or persistent LSIL/CIN 1 who met the treatment criteria, underwent cervical conization and were subsequently enrolled in the study. Patients who attended follow-up visits signed the informed consent approved by the local Ethics Commission (24/09/20; 19/320-E)”. (l. 125 – 128).

Point 8. Lines 136-138: If women were returned to the cervical cancer screening program after 3 years of negative follow-up results, were any women enrolled more than once? If so, how many? The authors should exclude the subsequent enrollments or use statistical methods that account for the collinearity in the data if women were enrolled more than once.

RE: Patients were only enrolled once. No women discharged to the screening program were referred back to our center. However, we cannot assure that patients have attended to another center. We have incorporated this information as a limitation of the study: “In addition, we cannot assure that women returned to screening program, presented late recurrence and were evaluated another hospital” (l. 418 – 420).

Point 9. Line 177: It should be univariate Cox regression analysis.

RE: I agree. The text has been modified as you propose (l. 201).

Point 10.  A flow diagram would clarify the disposition of participants. It is very difficult to follow as written in the text of results. I would also like to see the number enrolled, vaccinated, and with outcomes of interest stratified by time period (2009-2014 and 2015-2019) in a flow diagram.

RE: Following his recommendation, a flow chart has been introduced as figure 2 (l. 228). Redundant text has been removed.

Point 11. There are a few instances where the text states Error! Reference source not found. These should be reconciled.

RE: Sorry for the mistake. The cross reference must have been deleted. I have written it manually so it doesn't happen anymore. Original lines:

  • The absence of reference alluded to “Table 1”. It is already corrected in l. 209.
  • The absence of reference alluded to “Table 2”. It is already corrected in l. 272.
  • The absence of reference alluded to “Figure 2”. It is already corrected in l. 297.
  • The absence of reference alluded to “Table 3”. It is already corrected in l. 303.
  • The absence of reference alluded to “Table 4”. It is already corrected in l. 340.

Point 12. Percentage in Table 1 for HPV genotype are incorrect. Also, are these among HPV positive women? I think the authors should include the category for HPV negative.

RE: Thank you for your input. A non-HR HPV row has been added to the table 1.

Clinical Characteristics

Vaccinated group (%)

(n= 277)

Non-vaccinated Group (%)

(n = 286)

p

HPV Genotype

16/18

Other HR HPV

Non-HR HPV

66 (23.8)

51 (18.4)

11

39 (13.6)

42 (14.6)

14

0.187

Point 13. Lines 241-242: Why is the denominator for negative pap smear 241 and instead 563?

RE: The denominator was 563, but this data is not of interest, so it has been removed to avoid confusion.

We hope that we have provided enough detail for you to reconsider publishing our manuscript, and we remain at your disposal for any further clarification you may require.

Best Regards,

                                                               Andrea Casajuana.